# Characterization, Antioxidant and Antitumor Activities of Oligosaccharides Isolated from *Evodia lepta* (Spreng) Merr. by Different Extraction Methods

**DOI:** 10.3390/antiox10111842

**Published:** 2021-11-20

**Authors:** Feng Xiong, Hui-Xian Liang, Zhi-Jing Zhang, Taifo Mahmud, Albert S. C. Chan, Xia Li, Wen-Jian Lan

**Affiliations:** 1School of Pharmaceutical Sciences, Sun Yat-Sen University, Guangzhou 510006, China; xiongf28@mail2.sysu.edu.cn (F.X.); lianghx53@mail.sysu.edu.cn (H.-X.L.); zhangzhj37@mail2.sysu.edu.cn (Z.-J.Z.); chenxz3@mail.sysu.edu.cn (A.S.C.C.); 2Department of Pharmaceutical Sciences, Oregon State University, Corvallis, OR 97331, USA; Taifo.Mahmud@oregonstate.edu; 3College of Chemistry and Bioengineering, Guilin University of Technology, Guiling 541004, China; biology754@163.com

**Keywords:** *Evodia lepta*, oligosaccharide, antioxidant, antitumor

## Abstract

*Evodia lepta* (*E. lepta*) is a traditional Chinese herbal medicine with various biological activities. One of the active components of this widely used medicinal plant is believed to be an oligosaccharide. The extraction yields, structural characteristics, antioxidant, and antitumor activities of four oligosaccharide extracts obtained by hot water extraction (HEO), ultrasound-assisted extraction (UEO), enzyme-assisted (EEO), and microwave-assisted extraction (MEO) were investigated. Matrix-assisted laser desorption/ionization-time of flight-mass spectrometry (MALDI-TOF-MS), X-ray diffraction (XRD), and Scanning electron microscopy (SEM) results indicated that the extraction methods had a difference on the molecular mass distribution, structure, and morphology of the EOs. In addition, HEO and MEO showed strong antioxidant activities, which might be related to their uronic acid and protein contents. More interestingly, MEO was more active toward MDA-MB-231 cells compared to other cells, and cell growth inhibition was proposed to occur through apoptosis. Overall, microwave-assisted extraction is a promising technique for the extraction of high quality EO.

## 1. Introduction

Oxidative stress is one of the key factors in tumor formation and progression. It may cause significant damage to cells, e.g., by breaking the DNA double-strand and/or causing mutations in proto-oncogenes or tumor suppressor genes. If damages due to oxidative stress are not repaired by the cells, they may lead to tumors [1,2]. Therefore, for self-protection cells need antioxidants to maintain the balance of oxidative metabolism. Antioxidants can function as cancer chemopreventives by protecting cells from oxidative stress. Among known naturally occurring antioxidants are oligosaccharides. This class of natural products has become the subject of investigations in recent years as they may have some advantages over other natural antioxidants, e.g., good biocompatibility and less side effects. Consequently, many oligosaccharides have been evaluated for their antioxidant activities and their potentials as cancer chemopreventives or antitumor drugs [3,4].

Oligosaccharides are a class of biomolecules that are abundant and widely distributed in nature. They can be obtained from biological materials by direct extraction, e.g., water bath, ultrasound-assisted, enzyme-assisted, and microwave-assisted extractions, or by degradation methods using acids or enzymes. In addition to their function as the main source of energy, some oligosaccharides may have other biological activities, such as antioxidant [5], antitumor [6], anti-hypertension [7], anti-inflammatory [8], and intestinal flora regulators [9]. These different biological activities seem to correlate with the compositions of the oligosaccharides, their molecular mass, degrees of polymerization (DP), chain structures, and other factors. On the other hand, different extraction methods may affect the type or composition of the oligosaccharide obtained [10,11]. For example, two distinct oligosaccharide extracts were obtained from the American Cranberry Pomace when prepared by two different extraction methods; enzyme and the microwave-assisted alkaline methods. Although both extracts had the same composition of monosaccharides, their ratios were different. In addition, the DP of oligosaccharides obtained by the microwave-assisted alkaline method was higher (7–10 monomers) than the oligosaccharides obtained by the enzymatic method (2–5 monomers) [12].

*Evodia lepta* (Spreng) Merr (*E. lepta*), which belongs to the family *Rutaceae*, is a traditional Chinese herbal medicine commonly found in the south of China. In addition to its medicinal use, it has a long history being used as food, such as in Guangdong herbal tea [13]. The fat-soluble components of *E. lepta* [14], e.g., volatile oil, alkaloids, flavonoids, and coumarins, showed good anti-inflammatory [15], anti-viral [16], anti-bacterial [17], and antitumor [18] activities. Interestingly, the total sugar content of *E. lepta* is more than 30%, with oligosaccharides appearing to be the main active ingredients. While different extraction methods may affect the composition of the oligosaccharides and the quality of the products, there are no reports in the literature on the relationship between different extraction methods and the biological activity of *E. lepta* oligosaccharides (EO).

Therefore, on the basis of the literature and optimization (response surface method) of microwave-assisted extraction parameters, *E. lepta* was extracted by different methods including hot water extraction (HEO), ultrasound-assisted extraction (UEO), enzyme-assisted extraction (EEO), and microwave-assisted extraction (MEO). The physicochemical properties, structural characteristics, and biological activities of the EOs were evaluated. The results provide a suitable and promising extraction method for *E. lepta* and are helpful in the development of EO-related products.

## 2. Materials and Methods

### 2.1. Materials and Reagents

*E. lepta* was picked from the wild in Dapu County, Meizhou City, China, identified by the authors (F.X., and W.L.) and stored in School of Pharmaceutical Sciences, Sun Yat-sen University. Cellulase (50,000 U/g) was purchased from Shanghai Yuanye Bio-Technology Co., Ltd. (Shanghai, China). 2,2′-Azino-bis (3-ethylbenzothiazoline-6-sulfonic acid) (ABTS) was purchased from Beyotime Biotechnology Co., Ltd. (Shanghai, China). 1,1-Diphenyl-2-picrylhydrazyl (DPPH) was purchased from Shanghai Macleans Biochemical Technology Co., Ltd. (Shanghai, China). F-12K, RPMI 1640, and DEME medium were purchased from Yongjin BIOTECH Co., Ltd. (Guangzhou, China). Fetal bovine serum (FBS) was purchased from Jiandaoshou Gene Technology Co., Ltd. (Guangzhou, China). A549, HCT-116, Sjsa-1 and MDA-MB-231 cells were obtained from the Laboratory of Pharmacology and Toxicology, School of Pharmaceutical Sciences, Sun Yat-sen University (Guangzhou, China). All other chemical reagents used were analytical grade.

### 2.2. Extraction Process

#### 2.2.1. Preparation of Raw Material

Freshly picked branches and leaves of *E. lepta* were washed, dried, and crushed. Ultrasound-assisted extraction (400 W, 1 h, 3 times, 5 mL/g) with 95% ethanol was used to remove fat-soluble components. After filtration, the residue was used as a sample for subsequent experiments.

#### 2.2.2. Single-Factor Experiment

Single-factor experiments were performed to investigate the effects of liquid-to-solid ratio (10, 20–50 mL/g), time (2, 4–10 min), and microwave power (140, 280–700 W) on the extraction yield of EO. Glucose was used as a standard product to obtain a standard curve of the sugar content. The standard curve equation was *y* = 0.0163*X*, *R*^2^ = 0.999. The curve had a good linear relationship when the sugar content was 20–100 μg. The extraction yield of EO was calculated by the above standard curve.

#### 2.2.3. Response Surface Methodology Experiment Design

The range of extraction variables were preliminarily determined through a single-factor experiment. The three-level Box–Behnken design (BBD) was performed using three independent variables (liquid-to-solid ratio, time, microwave power). The variables were coded according to Equation (1):(1)$$x_{i}=\frac{X_{i}-X_{0}}{\Delta X_{i}}$$
where *χ_i_* is a coded value of a variable; *X_i_* is the actual value of a variable; *X*_0_ is the actual value of *X_i_* at the center point; and ∆*X_i_* is the step change value. Table 1 shows the range of independent variables and their levels. The independent variables and their ranges were determined in our preliminary experiments. Three experiments for each condition were performed and the mean values were considered as observed responses. The complete design was consisted of 17 experimental points, and the experiments were performed in a random order.

Data from the BBD were analyzed by multiple regressions to fit the following quadratic polynomial Equation (2):(2)$$Y=\beta_{0}+\sum_{i=1}^{k}\beta_{i}X_{i}+\sum_{i=1}^{k}\beta_{ii}X_{i}^{2}+\sum_{i=1}^{k-1}\sum_{j>i}^{k}\beta_{ij}X_{i}X_{j}$$
where *Y* represents the response function. *β*_0_ is an intercept. *β_i_*, *β_ii_*, and *β_ij_* are the coefficients of the linear, quadratic, and interactive terms, respectively. *X**_i_* and *X_j_* represent the coded independent variables. The fitted polynomial equation is expressed as surface and contour plots to visualize the relationship between the response and experimental levels of each factor and to deduce the optimum conditions. The regression coefficients of individual linear, quadratic, and interaction terms were determined through analysis of variance.

#### 2.2.4. Extraction Oligosaccharides with Different Methods

The above residue was subjected to four different extraction techniques: (1) the hot water method (90 °C, 120 min, 28 mL/g), (2) the ultrasound-assisted method (50 °C, 30 min, 28 mL/g, 400 W), (3) the enzyme-assisted method (cellulase, 50 °C, 30 min, 28 mL/g), and (4) the microwave-assisted method (6 min, 630 W, 28 mL/g). After completion of the extraction process, the solution obtained from the enzyme-assisted method was placed in boiling water (100 °C) for 10 min to inactivate the enzyme. The above extraction solutions were filtered, centrifuged, and concentrated, and then anhydrous ethanol (1:4, *v*/*v*) was added. The solution was allowed to stand at 4 °C overnight. The precipitate was reconstituted, and the solution was subjected to AB-8 resin column chromatography (26 mm × 40 cm). The above solution treated by AB-8 resin were concentrated and lyophilized to obtain the hot water extracted oligosaccharides (HEO), ultrasonic-assisted extracted oligosaccharides (UEO), enzyme-assisted extracted oligosaccharides (EEO), and microwave-assisted extracted oligosaccharides (MEO).

### 2.3. Physicochemical Properties

The content of the total sugars, reducing sugars, proteins, uronic acid, total flavonoids, and total phenols in the oligosaccharide samples were determined by the phenol-sulfuric acid method [19], 3,5-dinitrosalicylic acid colorimetry (DNS) method [20], Lowery method [21], m-hydroxydiphenyl method [22], aluminum nitrate colorimetric method [23], and the Folin–Denis method [24], respectively.

### 2.4. Molecular Weight Distribution of the Oligosaccharides

The molecular weight of the oligosaccharides was determined according to the reference method, with slight modifications [25]. The oligosaccharide solution (10 μL, 1 mg/mL) was mixed with 1-(4-cyanophenyl)-4-piperidinyl hydrazide (CPH) solution (40 μL) and the acetic acid solution (10 μL, absolute ethanol as the solvent, *v*/*v* = 0.125%) in a 1.5 mL Eppendorf tube and the mixture was placed in the water bath at 90 °C until the solution was dry. Subsequently, 50% aqueous acetonitrile (100 μL) was used to dissolve the derivatized sample, and the solution was analyzed by Matrix-assisted laser desorption/ionization-time of flight-mass spectrometry (MALDI-TOF-MS).

### 2.5. Fourier Transform Infrared Spectrometer Analysis

The dried oligosaccharide sample (3 mg) was added to dry potassium bromide (KBr) (100 mg) and the mixture was ground uniformly, compressed, and scanned at a wavelength range of 4000–400 cm^−1^ on a Fourier Transform Infrared Spectrometer (FT-IR). An empty KBr tablet was used as a blank background.

### 2.6. X-ray Diffraction Analysis

The oligosaccharide sample slide was prepared and placed on the sample stage. X-ray diffraction (XRD) analysis was performed by XPert3 Power (PANalytical B.V, Almelo, Nederland) type multifunction under Cu target with λ = 1.54056 A, 2*θ* range of 5° to 90°, and rate of 0.6565 °/s.

### 2.7. Scanning Electron Microscope Analysis

A small amount of the oligosaccharide was placed on a sample table contained conductive paste by spraying for 30 s. The images were taken by a HITACHI SU5000 Scanning electron microscope (SEM, Hitachi High-Tech Group, Tokyo, Japan) at a voltage of 5.0 kV and 1000 times magnification.

### 2.8. Antioxidant Activity In Vitro

#### 2.8.1. DPPH Radical Scavenging Assay

The oligosaccharide solution (2 mL) was mixed with 2 mL of DPPH solution (0.04 mg/mL, 95% ethanol as the solvent). The mixture was allowed to equilibrate for 30 min at room temperature in the dark, and then the absorbance (*Ai*) was measured at 517 nm. Additionally, the absorbance (*Ac*) of the DPPH ethanol solution (2 mL) + ethanol (2 mL) and the absorbance (*Aj*) of the oligosaccharide sample solution (2 mL) + ethanol (2 mL) was measured as described above. In this study, ascorbic acid (VC) was used as the positive control. The DPPH radical scavenging rate, which was defined as *K* (%) = [1 − (*Ai* − *Aj*)/*Ac*] × 100, was calculated [26].

#### 2.8.2. Hydroxyl Radical Scavenging Assay

The oligosaccharide solution (1 mL) was added to a mixture of FeSO_4_ solution (1 mL, 9 mmol/L) and salicylic acid solution (1 mL, 9 mmol/L, absolute ethanol as solvent). Subsequently, H_2_O_2_ solution (1 mL, 8.8 mmol/L) was added to start the reaction at 37 °C for 30 min. The absorbance of the oligosaccharides measured at 510 nm was designated as A1. The absorbance of distilled water (instead of oligosaccharide) was designated as A2, and the absorbance of distilled water (instead of H_2_O_2_) was designated as A3. VC was used as a positive control. The hydroxyl radical scavenging rate, which was defined as *K* (%) = [*A*2 − (*A*1 − *A*3)]/*A*2 × 100, was calculated [27].

#### 2.8.3. ABTS Radical Scavenging Assay

The ABTS working solution was prepared according to the kit instructions. In a 96-well plate, the ABTS working solution (200 μL) was mixed with the oligosaccharide solution (10 μL). The mixture was incubated at room temperature for 6 min, and the absorbance (A1) at 734 nm was measured. The absorbance of distilled water (instead of the oligosaccharide solution) was designated at A2. The absorbance of distilled water (instead of the ABTS working solution) was designated as A3. VC was used as a positive control. The ABTS radical scavenging rate, which was defined as *K* (%) = [*A*2 − (*A*1 − *A*3)]/*A*2 × 100, was calculated [28].

### 2.9. Antitumor Activity In Vitro

#### 2.9.1. Cell Culture

A549 cell was cultured in Kaighn’s Modification of Ham’s F-12 (F-12K) medium with 10% FBS. HCT-116 and Sjsa-1 cells were cultured in Roswell Park Memorial Institute 1640 (RPMI 1640) medium containing 10% FBS. MDA-MB-231 cell was cultured in Dulbecco’s modified eagle medium (DMEM) containing 20% FBS. Then, 1% antibiotic (penicillin/streptomycin) was added to the above-mentioned medium. All cell lines were incubated at 37 °C under 5% CO_2_. The A549, HCT-116, Sjsa-1, and MDA-MB-231 cells were used to determine the antitumor activity of EOs.

#### 2.9.2. Cell Proliferation Assay

The cells in the logarithmic growth phase were digested to prepare a cell suspension and counted. The A549, HCT-116, Sjsa-1 and MDA-MB-231 cells were transferred to 96-well plates at 5 × 10^5^ cells per well. After culturing for 24 h, the medium was replaced with fresh medium containing different concentrations of HOE, UOE, EOE, and MOE (0, 100, 200, 300, 400, 500, 600 μg/mL). After culturing for another 24 h, The 10 μL of Cell Counting Kit-8 (CCK-8, Dongren Chemical Technology (Shanghai) Co., Ltd, Shanghai, China) was added to each well and incubated for 1 h. The OD value of cells at 450 nm was measured by the Multimode Microplate Reader. The cell survival rate of each group was calculated according to the following formula: Cell viability (%) = *A_sample_*/*A_control_* × 100%, *A_sample_* was the average absorbance of the tested group, *A_control_* was the average absorbance of the control group.

#### 2.9.3. Analysis of MDA-MB-231 Cells Apoptosis

The MDA-MB-231 cells in the logarithmic growth phase were suspended in DMEM medium and transferred into 6-well culture plates with 5 × 10^5^ cells per well. After 24 h incubation, the medium was replaced with a fresh medium containing MPE (0, 300, 600 μg/mL). When the cell coverage was about 70%, the cells were digested, centrifuged, and washed with a pre-cooled PBS buffer at 4 °C. The cells were stained with 10 μL Annexin V- Fluorescein Isothiocyanate (FITC)/Propidine iodide (PI), incubated in the dark at room temperature for 10–15 min, and tested on the Flow Cytometer (CytoFLEX, Beckman Coulter Inc., Brea, CA, USA).

### 2.10. Statistical Analysis

Experimental data were expressed as mean ± SD. The Design-Expert software version 12.0 (Stat-Ease Inc., Minneapolis, MN, USA) was used for the experimental design. *P* values of less than 0.05 indicate statistical significance. Graphs, calculation of IC_50_ values, and analysis of variance were obtained or performed using GraphPad Prism 8.02 (GraphPad Software Inc., San Diego, CA, USA).

## 3. Results and Discussion

### 3.1. Single-Factor Experiments

The liquid-to-solid ratio (10–50 mL/g), time (2–10 min), and microwave power (140–700 W) were used to investigate the effects of those parameters on the extraction yield of oligosaccharides. The effect of the liquid-to-solid ratio on the extraction yield of EO is shown in Figure 1A. Following the change in the liquid-to-solid ratio, the extraction yield of EO increased. It reached the highest point of 4.92% at the liquid-to-solid ratio of 20 mL/g. However, when the liquid-to-solid ratio was further increased, the extraction yield of EO decreased. That may be the long diffusion distance of the internal tissue and mass transfer loss during transmission process. The effect of microwave time on the extraction yield of EO is shown in Figure 1B. Similar to the effect of the liquid-to-solid ratio, the extraction yield improved following the extension of the extraction time. It reached the highest extraction yield when the extraction time was set at 4 min, then it showed a downward trend with the extension of time to over 4 min. The effect of microwave power on the extraction yield of EO is shown in Figure 1C. The maximum extraction yield of 4.86% was reached when the microwave power was set at 560 W. Increasing the microwave power beyond 560 W resulted in decreased oligosaccharide extraction yield.

### 3.2. Response Surface Analysis

Next, using the above optimized parameters (liquid-to-solid ratio 20 mL/g, time 4 min, and microwave power 560 W) as the initial values, we performed a response surface analysis. The liquid-to-solid ratio (*X*_1_), time (*X*_2_), and microwave power (*X*_3_) were used as factors, and the extraction yield of EO was used as the response value *Y*. The experimental conditions were designed by BBD and the experimental data are shown in Table 2.

The extraction yields of EO were 3.84%~5.07%, and the quadratic polynomial regression was fitted to the experimental data to obtain the regression equation:*Y* = −0.29*X*_1_^2^ − 0.20*X*_2_^2^ − 0.08*X*_3_^2^ + 0.31*X*_1_*X*_2_ + 0.09*X*_1_*X*_3_ − 0.16*X*_2_*X*_3_ + 0.24*X*_1_ + 0.24*X*_2_ + 0.22*X*_3_ + 4.7(3)

The analysis of variance (ANOVA) of the results is summarized in Table 3. The model *p*-value (<0.01) suggested that the regression model is significant. The regression equation model correlation coefficient (*R*^2^) and the adjusted model correction coefficient (Adj *R*^2^) were 0.96 and 0.92, respectively, indicating that the equation model in this experiment is highly significant, small experimental errors, and high reliability. The primary term (*X*_1_, *X*_2_ and *X*_3_) and the interactive term (*X*_1_*X*_2_ and *X*_2_*X*_3_) were significant factors (*p* < 0.05). It can be seen that the liquid-to-solid ratio, the time, and the microwave power have significant effects on the yield of EO (*p* < 0.01). The order of the effect of each factor on the yield of EO was time (*X*_2_), liquid-to-solid ratio (*X*_1_), and microwave power (*X*_3_).

Based on the regression model, the response surface contour lines of the interactions between the liquid-to-solid ratio, time, and microwave power are shown in Figure 2. The coefficients of the quadratic terms (*X*_1_^2^, *X*_2_^2^, *X*_3_^2^) in the equation were all negative values, and the equation parabolic opening downward had a maximum value [29], which was the highest point at the response surface. The interaction between the liquid-to-solid ratio and the microwave power at an extraction time of 4 min is shown in Figure 2A. The microwave power was constant, the yield displayed a trend of first increasing and then decreasing. The yield increased with increasing microwave power at the fixed extraction time. The interaction between the liquid-to-solid ratio and microwave power was not significant (*p* > 0.05). The interaction between the liquid-to-solid ratio and time with a fixed microwave power of 560 W is shown in Figure 2B. With increasing liquid-to-solid ratio and time, yield continued to increase and the interaction was stronger (*p* < 0.01). The interaction between microwave power and time while the liquid-to-solid was 20 mL/g is shown in Figure 2C. The extraction time had less of an impact on the yield at a higher power, and microwave power had less of an effect on yield at a longer time. Therefore, the interaction was relatively weak (*p* < 0.05).

### 3.3. Verification of the Predictive Model

On the basis of the response surface analysis, the optimum extraction parameters for microwave-assisted extraction of EO are as follows: liquid-to-solid ratio, 27.85 mL/g; extraction time, 5.99 min; and microwave power, 658.94 W. The theoretical maximum yield of these parameters is 5.10%. Combined with the feasibility of experimental equipment, the optimized parameters were adjusted to liquid-to-solid ratio of 28 g/mL, extraction time of 6 min, and microwave power of 630 W. The extraction yield under these conditions was 4.90 ± 0.05%. The theoretical value (4.92%) was basically consistent with the actual value, indicating that the process conditions were stable and feasible.

### 3.4. Physicochemical Properties of the Oligosaccharides

EOs were extracted by hot-water, ultrasound-assisted, enzyme-assisted, and microwave-assisted extraction methods to obtain HEO, UEO, EEO, and MEO, respectively. Due to the decolorization of AB-8 resin chromatography column, HEO, UEO, EEO, and MEO were all obtained as gray powders. The primary physicochemical properties of EOs are shown in Table 4. The data also show that the oligosaccharides yields were affected by the extraction techniques (*p* < 0.05). The yields of the four oligosaccharide samples were determined to be 3.58% for HEO, 4.80% for UEO, 4.90% for MEO, and 5.30% for EEO. This suggests that ultrasound-assisted, microwave-assisted, and enzyme-assisted extraction technology could significantly increase the extraction yields of oligosaccharides. The content of total sugar in EOs were not affected by the extraction methods. However, the uronic acid and protein content of HEO and MEO, UEO and EEO were significantly different (*p* > 0.05). The uronic acid content of HEO and MEO were 5.00% and 5.10%, respectively, which were higher than that of UEO and EEO. According to the literature, there is good correlation between the content of uronic acid and antioxidant activity [30]. Therefore, HEO and MEO are expected to have greater biological activities than UEO and EEO.

### 3.5. Molecular Weight Distribution Analysis

Oligosaccharides are not easily protonated in MALDI-TOF MS because of the lack of basic sites. Instead, they tend to form alkali metal associated ions [31]. In the positive ion mode mass spectrometry, oligosaccharides are mainly observed as their sodiated and potassiated ions, having a characteristic cluster of ions that differ by 16 Da [32], which is consistent with the difference between the atomic masses of Na and K. Oligosaccharides obtained from the above extraction methods also conform to that fact. To avoid the measurement results being influenced by the matrix (CPH), small molecules below 500 Da were deflected in the MALDI-TOF-MS experiment. As shown in Figure 3, the molecular weight of EOs were in the range of 500–1500 Da, and the degree of polymerization were 3–8. Corresponding to the abundance of oligosaccharides with different DPs, the putative tetrasaccharide showed the highest intensity. Under the same DP, the EOs all showed several groups of different characteristic ions peaks, which were presumed to be hetero-oligosaccharides composed of a variety of different monosaccharides. Combining Figure 3E,F, HEO had a wider range of molecular weight distribution, indicating that different extraction methods could affect the molecular weight distribution. High temperatures and extended extraction times have been reported to cause the aggregation of oligosaccharide, which increased the molecular weight, and DP [33]. Among these methods, HEO had the largest DP. It was reported that Li et al. used hot water, ultrasound-assisted, and enzyme-assisted methods to prepare mulberry fruit polysaccharides, and the molecular weight of the hot water extract was the largest, which was consistent with the results of this study [34].

### 3.6. FT-IR Analysis

The FT-IR spectra of HEO, UEO, EEO and MEO showed that all of them share the same characteristic functional groups (Figure 4). Three absorption peaks caused by O-H stretching vibration (3418–3442 cm^−1^), C-H stretching vibration (2931–2948 cm^−1^), and C-H variable angle vibration (1413–1431 cm^−1^) were consistent with those expected for saccharides [35]. The absorption peaks at 1600–1650 cm^−1^ caused by C=O asymmetric vibration in -COO, indicated that EOs contain uronic acid [36]. The absorption peaks at 1000–1100 cm^−1^ belong to C-O stretching vibration, and those at 891 ± 7 cm^−1^ belong to β-configuration. It is concluded that the various extraction methods described above did not affect the type of oligosaccharides being extracted, at least not those related to their glycosidic bonds and conformations.

### 3.7. XRD Analysis

To determine the physical properties of the EO samples, XRD analysis was performed (Figure 5). The peak shape trends of 2*θ* were similar in the range of 5° to 90°, and both weak and broad diffraction peaks appeared at around 20°, which showed low crystallinity [37]. These findings indicated that the interiors of the four oligosaccharides have an amorphous structure. Based on Bragg’s equation for crystal diffraction (2d sin *θ* = nλ), the lattice spacing of HEO, UEO, EEO and MEO were 4.39 nm, 4.35 nm, 4.42 nm, and 4.37 nm at *n* = 1 and *λ* = 1.54 Å. The results showed that EOs obtained by hot-water, ultrasound-assisted, enzyme-assisted, and microwave-assisted methods had similar effects on the internal crystallinity, which led to similar powder states of the four oligosaccharide samples. While the 2*θ* of the diffraction peak of MEO was similar to that of other samples, the peak intensity was significantly lower, indicating that microwave-assisted extraction is more likely to affect the properties of the crystals.

### 3.8. SEM Analysis

SEM is an important technique to characterize the microscopic morphological characteristics of polymer materials. To investigate whether different extraction methods could affect the morphology of the resulted EOs, SEM images of the four EO samples were obtained (Figure 6). The results showed that HEO formed a flake-shaped with cracks and holes on the surface. UEO presented a porous sheet shape with smaller holes on the surface compared to HEO. EEO had a flat sheet shape without holes. MEO has a porous and wavy surface. In contrast to EEO, the surface of UEO and MEO has many pores, which may be due to ultrasonic cavitation or microwave radiation acting on glycosidic bonds, respectively. Under the assistance of ultrasounds, enzymes, or microwaves, the intermolecular hydrogen bonds in HEO may be broken, resulting in decreased chain conformational stability and intermolecular cross-linking [38]. This may lead to a decrease of surface flatness and hole diameter.

### 3.9. Antioxidant Activities In Vitro

To evaluate the antioxidant activity of EOs, DPPH, ABTS, and hydroxyl radical scavenging assays were used to measure the free radical scavenging activity. In the DPPH assay, HEO, UEO, EEO, and MEO showed concentration-dependent scavenging activities with IC_50_ values of 0.11 mg/mL, 0.16 mg/mL, 0.17 mg/mL, and 0.15 mg/mL, respectively (Figure 7A). In the hydroxyl radical scavenging assay, all EOs also showed concentration-dependent scavenging activities with IC_50_ values of 1.96 mg/mL, 1.95 mg/mL, 1.56 mg/mL, and 1.21 mg/mL for HEO, UEO, EEO, and MEO, respectively (Figure 7B). Similarly, all of the samples showed concentration-dependent scavenging activities in the ABTS assay (Figure 7C). The IC_50_ values for HEO, UEO, EEO, and MEO were 0.74 mg/mL, 0.92 mg/mL, 1.69 mg/mL, and 0.98 mg/mL, respectively. While EOs showed some antioxidant activity, they were not as strong as VC with an IC_50_ of 0.02 mg/mL. Among the oligosaccharide samples, HEO showed the highest radical scavenging activity in the DPPH and ABTS assays (*p* < 0.05). However, interestingly, in the hydroxyl radical scavenging assay, MEO turned out to be the most active sample (*p* < 0.01).

The free radical scavenging activity of oligosaccharides is believed to be affected by many factors. One of them appears to be the presence of uronic acid. It has been reported that the content of uronic acid in oligosaccharides is closely related to their biological activity. The presence of uronic acid induces saccharides to be negatively charged [39]. A higher uronic acid content will lead to a greater absolute value of the Zeta potential, which in turn may result in stronger antioxidant activity [34]. In addition, some studies have shown that protein content was one of the factors that affected the antioxidant capacity of oligosaccharides, as -NH_2_ can absorb hydrogen ions in the solution to form-NH_3_^+^ and reacts with the radicals [40]. Compared to UEO and EEO, both HEO and MEO had higher contents of uronic acid and protein, which may explain why HEO and MEO exhibited superior antioxidant activities. Oxidative stress is one of the key factors in tumor progression. The use of antioxidants to protect cells from oxidative stress is an important strategy for tumor treatment, higher antioxidant activity will have the better inhibitory effects for tumor cells to a certain extent. Therefore, HEO and MEO may maintain a high inhibition rate of tumor cells proliferation.

### 3.10. Cells Proliferation Activity

The relationship between antioxidant activity and antitumor activity of EOs under different methods was analyzed. The different concentrations (0–600 μg/mL) of EOs were used to treat A549, HCT-116, Sjsa-1, and MDA-MB-231 tumor cells. After 24 h incubation, cell proliferation inhibitory activity was analyzed by CCK-8. The results showed that EOs could inhibit Sjsa-1 and MDA-MB-231 cell growth in a dose-dependent manner, but had no obvious effects on the viability of A549 and HCT-116 cells (Figure 8). It appears that MDA-MB-231 cells are more sensitive to EOs compared to Sjsa-1 cells. The survival rates of MDA-MB-231 cells treated with 600 μg/mL of HEO, UEO, EEO and MEO were 63.78%, 64.34%, 42.59% and 36.53%, respectively. Additionally, the inhibitory effect of MEO was significantly higher than that of HEO. Interestingly, MEO was more active than the other EOs in the hydroxyl radical scavenging assay, but HEO showed the strongest antioxidant activity in the DPPH and ABTS radical scavenging assays, which indicates that the antitumor activity of EOs is not only related to their antioxidant capacity, but may also be related to their surface structure under different methods. Compared with HEO, the porous structure of MEO could increase its specific surface area, leading to increasing the probability of contact with tumor cell surface sites, which may enhance the tumor cell inhibitory effect of EOs. Since the effect of MEO on MDA-MB-231 cells was more prominent than other samples, MEO and MDA-MB-231 cells were selected for further studies.

### 3.11. Analysis of Apoptosis by Flow Cytometry

To further investigate the effect of MEO on cell growth, MDA-MB-231 tumor cells were incubated with different concentrations of MEO for 24 h, and stained with Annexin V-FITC/PI for flow cytometry analysis. The results showed that the proportions of apoptotic cells in cultures incubated with 300 and 600 μg/mL of MEO were 17.11% and 20.28%, respectively (Figure 9). Interestingly, the number of early-stage apoptotic cells in samples treated with 600 μg/mL of MEO was lower than those treated with 300 μg/mL of MEO. However, the trend reversed for late-stage apoptotic cells. The above results suggest that MEO can cause cancer cell death by inducing apoptosis.

## 4. Conclusions

In this study, we investigated the effects of different extraction methods (hot water, ultrasound-assisted, enzymatic, and microwave-assisted) on the physicochemical properties, structural characteristics, antioxidant, and antitumor activities of EOs. Optimization of the microwave-assisted extraction conditions by response surface methodology provided ideal parameters (liquid-to-solid, 28 mL/g; time, 6 min; and microwave power, 630 W) with an extraction yield of 4.90 ± 0.05%. The molecular weight distribution range of EOs extracted by different methods were 500–1500 Da, with EEO having the highest yield, and there being significant differences in the biological activities of EOs from different extraction methods. Interestingly, MEO was the most active among the EOs in inhibiting the growth of MDA-MB-231 cells, most likely by inducing apoptosis. Our study suggests that, among the four methods tested, the microwave-assisted method is the best extraction method for EOs based on the oligosaccharide yield and their strong antioxidant and antitumor activities. Further studies on the MEO sample are currently being pursued in our laboratory. These include characterization of the chemical structures of the oligosaccharides, more in-depth investigation of its in vitro and in vivo biological activities, and its mechanism of action.

## Figures and Tables

**Figure 1 antioxidants-10-01842-f001:**
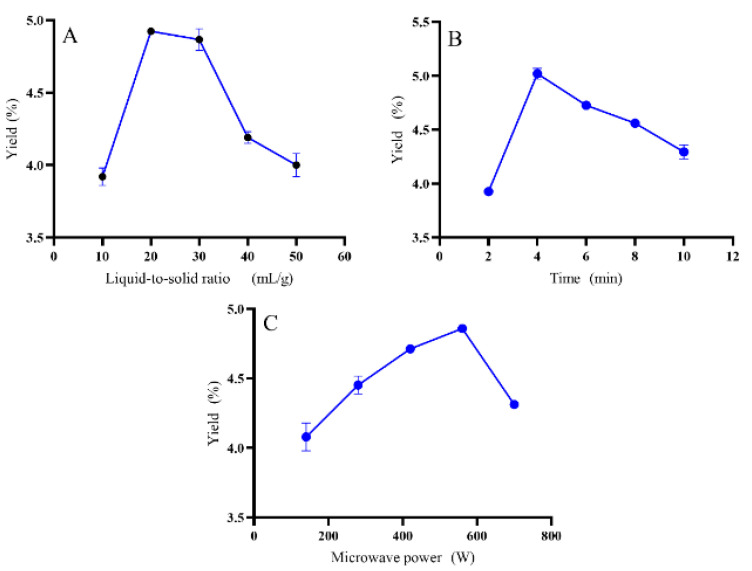
Effects of various factors on the yield of EO, (**A**) is the liquid-to-solid ratio factor, (**B**) is the time factor, (**C**) is the microwave power factor.

**Figure 2 antioxidants-10-01842-f002:**
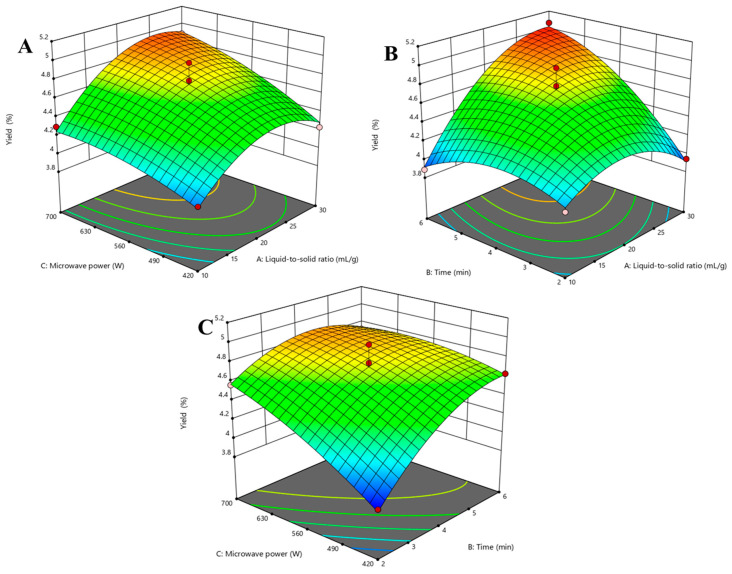
Response plots of the effects of various factors on the yield of EO. (**A**) is the graph of interaction between the liquid-to-solid ratio and microwave power; (**B**) is the graph of interaction between the liquid-to-solid ratio and time; (**C**) is the graph of interaction between l microwave power, and time.

**Figure 3 antioxidants-10-01842-f003:**
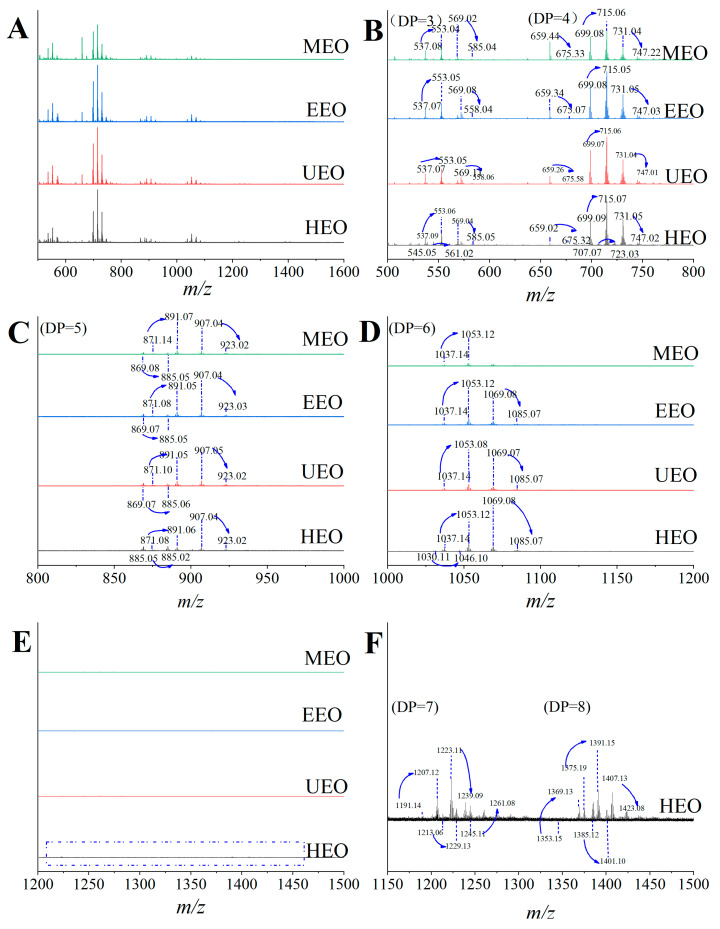
Molecular weight distribution analysis of EOs; (**A**) is the partial mass spectra of EOs from different extraction methods, (**B**) is the magnification of partial mass spectra of EOs between *m*/*z* 500 and 800, (**C**) is the magnification of partial mass spectra of EOs between *m*/*z* 800 and 1000, (**D**) is the magnification of partial mass spectra of EOs between *m*/*z* 1000 and 1200, (**E**) is the magnification of partial mass spectra of EOs between *m*/*z* 1200 and 1500, and (**F**) is the magnification of partial mass spectra of HEO between *m*/*z* 1180 and 1450.

**Figure 4 antioxidants-10-01842-f004:**
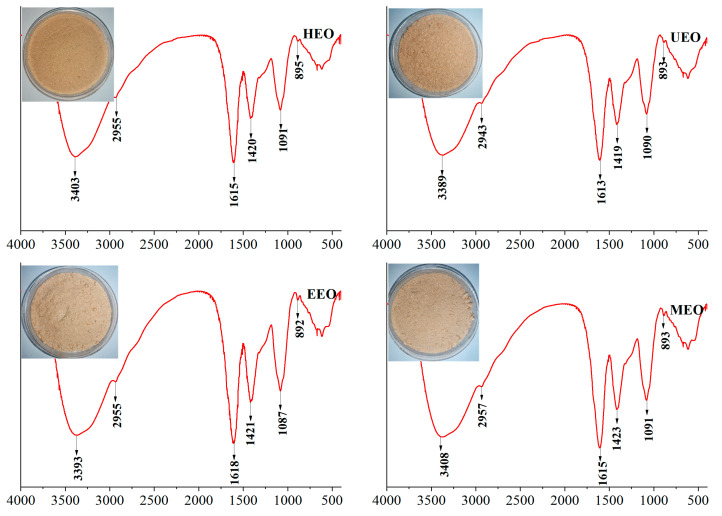
FT-IR (Fourier Transform Infrared Spectrometer) chromatograms of EOs.

**Figure 5 antioxidants-10-01842-f005:**
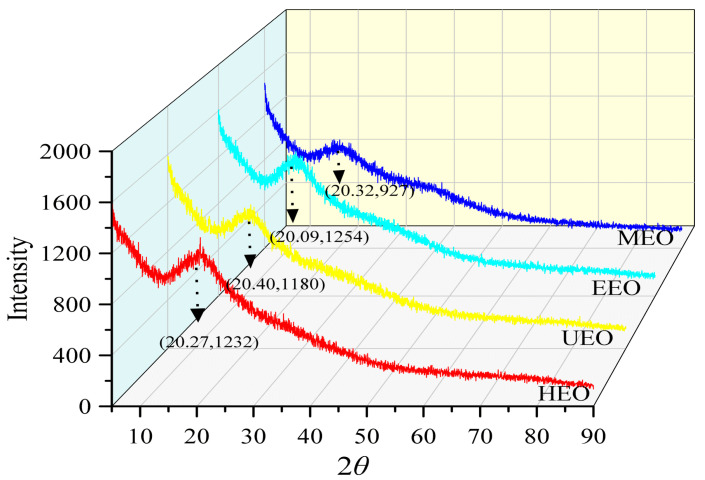
XRD (X-ray diffraction) pattern analysis of EOs.

**Figure 6 antioxidants-10-01842-f006:**
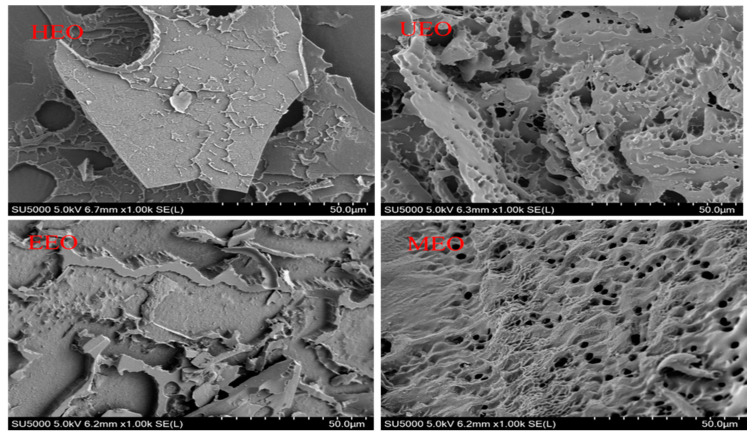
SEM (scanning electron microscopy) image analysis of EOs.

**Figure 7 antioxidants-10-01842-f007:**
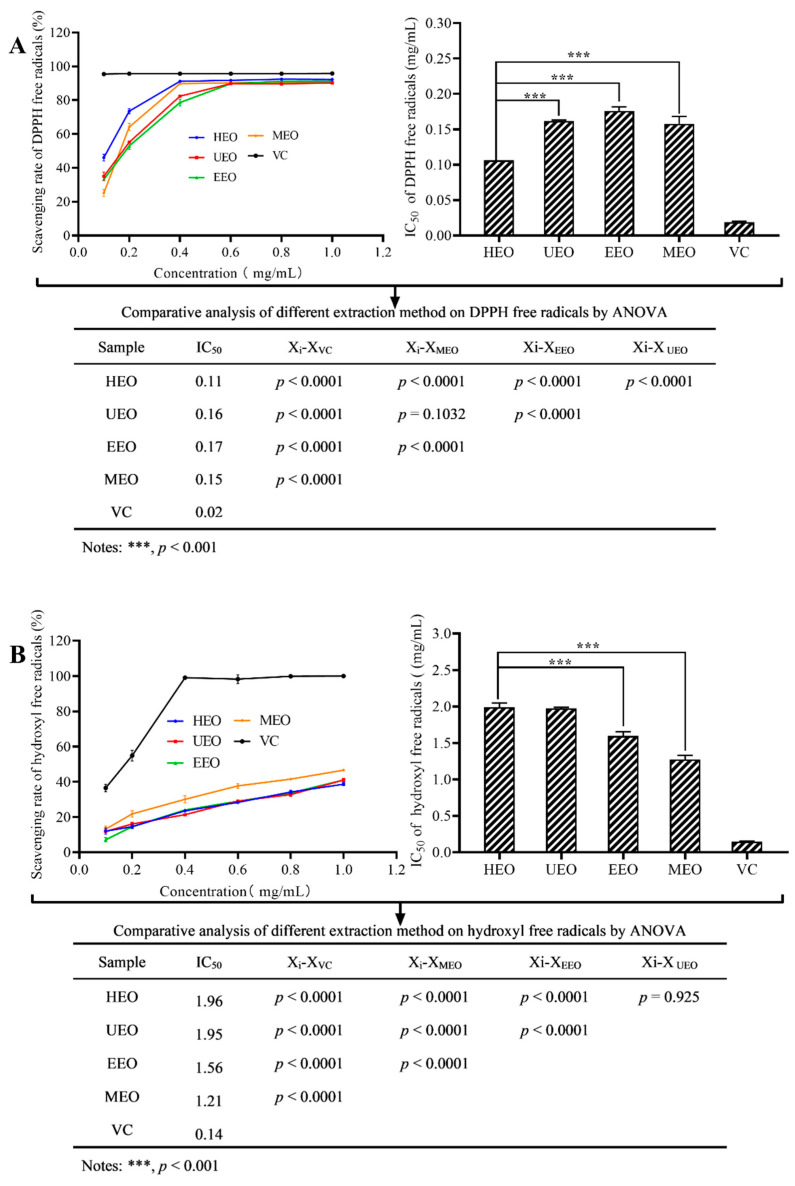
Free radical scavenging activities of EOs, (**A**) is DPPH free radicals, (**B**) is hydroxyl free radicals, (**C**) is ABTS free radicals.

**Figure 8 antioxidants-10-01842-f008:**
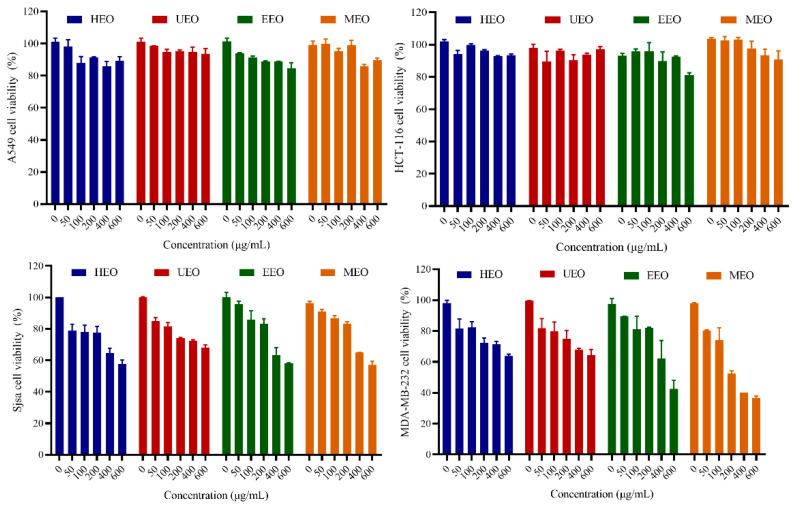
Proliferation activity of EOs on cancer cell lines.

**Figure 9 antioxidants-10-01842-f009:**
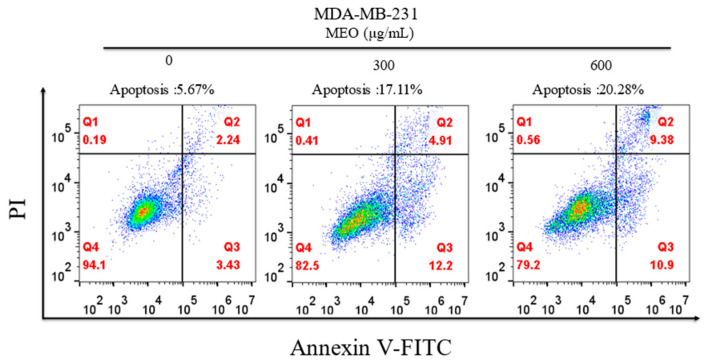
Flow cytometry analysis of MDA-MB-231 cells treated with MEO.

**Table 1 antioxidants-10-01842-t001:** Independent variables and their levels in the response surface design.

Independent Variables	Factor Level		
	−1	0	1
*X*_1_: Liquid-to-solid ratio (mL/g)	10	20	30
*X*_2_: Time (min)	2	4	6
*X*_3_: Microwave power (W)	420	560	700

**Table 2 antioxidants-10-01842-t002:** Design and results of response surface methodology.

Experiment Number	Liquid-to-Solid Ratio (mL/g)	Time (min)	Microwave Power (W)	Yield (%)
1	20	4	560	4.66
2	30	2	560	3.94
3	20	4	560	4.77
4	30	4	700	4.89
5	30	4	420	4.23
6	20	6	420	4.63
7	20	4	560	4.57
8	20	4	560	4.68
9	10	6	560	3.89
10	20	4	560	4.95
11	10	4	420	4.00
12	10	2	560	4.01
13	20	2	700	4.56
14	20	2	420	3.84
15	30	6	560	5.07
16	10	4	700	4.30
17	20	6	700	4.72

**Table 3 antioxidants-10-01842-t003:** Regression model variance analysis results.

Source	Sum of Squares	DF	Mean Square	*F* Value	*p*-Value	Significance Level
Model	2.471	9	0.275	19.507	0.00001	**
*X* _1_	0.467	1	0.467	33.17	0.001	**
*X* _2_	0.477	1	0.477	33.884	0.001	**
*X* _3_	0.393	1	0.393	27.904	0.001	**
*X* _1_ *X* _2_	0.386	1	0.386	27.434	0.001	**
*X* _1_ *X* _3_	0.032	1	0.032	2.25	0.177	
*X* _2_ *X* _3_	0.1	1	0.1	7.095	0.032	*
*X* _1_ ^2^	0.357	1	0.357	25.37	0.002	**
*X* _2_ ^2^	0.182	1	0.182	12.928	0.009	**
*X* _3_ ^2^	0.027	1	0.027	1.924	0.208	
Residual	0.099	7	0.014			
Lack of Fit	0.013	3	0.004	0.206	0.887	
Pure Error	0.085	4	0.021			
*R*^2^ = 0.96			Adj *R*^2^ = 0.92			

Notes: *, *p* < 0.05; **, *p* < 0.01; DF: Degree of Freedom; Adj *R*^2^: Adjusted model correction coefficient.

**Table 4 antioxidants-10-01842-t004:** Primary physicochemical properties of EOs.

Sample	Total Sugar (%)	Reducing Sugar (%)	Protein (%)	Uronic Acid (%)	Total Phenol (%)	Flavonoid (%)	Extraction Rate (%)
HEO	36.59 ± 1.07	9.76 ± 0.80	12.98 ± 0.15	5.00 ± 0.04	5.84 ± 0.02	6.83 ± 0.25	3.58 ± 0.04
UEO	37.23 ± 0.97	9.96 ± 0.82	9.98 ± 0.48 **	4.79 ± 0.02 **	6.35 ± 0.04 **	9.21 ± 0.10 **	4.80 ± 0.11 **
EEO	32.68 ± 0.67 *	9.16 ± 0.46	10.76 ± 0.15 **	4.54 ± 0.10 **	4.58 ± 0.43	6.33 ± 0.05	5.30 ± 0.06 **
MEO	37.07 ± 0.15	9.56 ± 0.48	11.40 ± 0.33	5.10 ± 0.14	5.77 ± 0.40	6.18 ± 0.12	4.90 ± 0.05 **

Notes: *, *p* < 0.1; **, *p* < 0.05 compared with the control group (HEO, hot water extraction). UEO: ultrasound-assisted extraction; EEO: enzyme-assisted; MEO: microwave-assisted extraction.

## Data Availability

Data is contained within the article.

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
