# Peer review of "Characterization, Antioxidant and Antitumor Activities of Oligosaccharides Isolated from Evodia lepta (Spreng) Merr. by Different Extraction Methods"

_antioxidants, 2021, doi:10.3390/antiox10111842_

Round 1

Reviewer 1 Report

The manuscript compares four methods of oligosaccharide extraction from Evodia lepta. The physico-chemical and structural properties of the obtained samples were analyzed, as well as their antioxidant and anticancer properties. The studies performed are described and documented in detail. The manuscript is interesting and quite well written, but there are some elements that need improvement.

Minor remarks

  1. The last two sentences of the Introduction are basically conclusions and should be deleted.
  2. In many places, letters or numbers indicating superscript or subscript are inappropriately written, this should be corrected. For example, comments on equations 1 and 2, H2O2, FeSO4, 5x105 cells per well, etc.
  3. Equation (2) - there are variables Xi, Xj in the equation, but χi and χj in the text. Which version is correct?
  4. The word eluents included in the sentence on line 141 should be changed to a more appropriate word. The word eluent describes the mixture of solvents used to elute compounds from the column, not the solution eluted from the column. By the way, was any eluent used in this case, if so, what kind?
  5. Line 155 - what was the ratio of ethyl acetale and acetic acid solution?
  6. Section 2.8.2 and beyond - does ascorbic acid have the abbreviation VC or Vc because interchangeably both are used?
  7. Section 3.3 - On the basis of the response surface analysis microwave power was 658.94 W. However after analysis it was decided to be 630 W, why?
  8. Item 3.4 - extraction field for MEO was 4.98% according to text, but 4.90% according to Table 4. Which value is correct?
  9. Line 419 - what does the number 34 mean? If this is a literature item, the bracket is missing.

Author Response

Dear Sir:

Thank you for the pertinent comments and valuable suggestions. According to the specific opinions of the reviewers, we have already revised the article. Thanks to the editor for giving us the opportunity to modify and explain this time. The more detailed modifications are shown in the attachment.

Reviewer 2 Report

The aim of the paper has been to compare the extraction method to obtain active oligosaccharides from E. lepta regarding to physical and chemical properties and the structural characteristics.

Generally the paper is well written and deserve to be published. There are some comments:

In the abstract the paragraph from L 16-L18 could be removed.

The aim of the paper should be more focussed.

Figure 3 is not clear.

Author Response

(The authors gave the same response as above.)

Reviewer 3 Report

The paper is quite clear and interesting, however the spectral images are too small and therefore difficult to analyze.
the same is true for images relating to cell proliferation activity.
in the activity assays in cells of tumor cell lines it would be interesting to verify the activity and/or toxicity in non-tumor cells

Author Response

(The authors gave the same response as above.)
